# Long-term brain pressure monitoring via a discrete microimplant; a first-in-human safety and initial efficacy trial in adults and children with hydrocephalus

Simon C. Malpas [1,2,4] ✉, Bryon E. Wright [1,2,4], Sarah-Jane Guild[1,2,4], Peter Heppner[3], Robert J. Gallichan[1,2], Dixon P. Leung[1,2], Sang Ho Kim[1], Quinn Boesley [1], Sheryl Tan [1], Masahiro Kondo[2], Davina J. McAllister[3], John A. Windsor [1,3], Doug Campbell[1,3], P. Alan Barber [1,3] & Daniel McCormick[1,2]

Emerging neurotechnologies such as brain-computer interfaces and implantable sensors offer considerable promise in the treatment of a broad range of neurological conditions. The key challenges are reducing the implant size, powering it, and confirming long-term accuracy and safety. Here we report the development of a novel type of implantable medical device that measures intracranial pressure long term and which weighs only 0.28 g. Currently the management of hydrocephalus patients relies heavily on non-specific symptoms e.g. headache and there is a lack of actionable data to drive decisions that are not solely hospital based such as imaging. The implant is designed to sit within the cerebral cortex. In a group of 10 adults and 10 children with hydrocephalus we demonstrated that the device was safe and capable of remotely monitoring intracranial pressure in patients at home for up to 18 months (ClinicalTrial.gov NCT06402786). In several children shunt failures occurred and these were associated with raised ICP. Instead of relying on non-specific symptoms such as headache, physicians were able to obtain real-time intracranial pressure readings that can lead to changes in the management of these complex patients.

The past decade has seen growing interest in the development of novel medical devices for treating neurological conditions. Brain-computer interfaces, nerve stimulators and sensors of neuronal activity are all under development or have been approved[1]. These devices share a common design featuring a deeply implanted electrode or sensor tethered to a larger, superficially implanted enclosure that houses the battery and electronics. This "tethered" design has led to substantial issues including electrode breakages, infection risk, the need for battery replacements and damage to neuronal tissue from electrode or catheter movement[2,3].

We present a new paradigm of implantable medical device, where the implant is kept as small as possible and can be effectively located anywhere in the body. It can be made capable of stimulating or sensing physiologically relevant signals and receiving wireless power from, as

[1]Auckland Bioengineering Institute and Faculty of Medicine and Health Sciences, University of Auckland, Auckland, New Zealand. [2]Kitea Health Ltd, Auckland, New Zealand. [3]Auckland City Hospital, Auckland, New Zealand. [4]These authors contributed equally: Simon C. Malpas, Bryon E. Wright, Sarah-Jane Guild. ✉e-mail: s.malpas@auckland.ac.nz

**Fig. 1 | Overview of the Implantable ICP Monitoring System and Sensor Architecture. A** Schematic of the overall system illustrating the placement of the sensor in the cortex, receiving the ICP signal via an external wand and transfer of data to an App and onto a clinical portal for remote viewing. The system reports both mean and waveform ICP. **B** An ICP sensor placed next to the proximal end of a ventricular shunt indicating the similarity in size to a shunt (scale bar 3.5 mm). **C** Sensor architecture and manufacturing processes.

well as wirelessly communicating to an external reader. The implant must be small enough not to damage tissue or distort physiology while hermetically isolating electronics and ensuring the accuracy and stability of the measurements or stimulation over an extended period (years).

## Results

Here we describe a long-term monitoring system, including an implant, that is focused on sensing intracranial pressure (ICP) (Fig. 1A). This is particularly relevant for people with hydrocephalus which is characterized by increased ICP and is usually treated by draining excess cerebral spinal fluid (CSF) via surgically implanted shunts. Such shunts have one of the highest failure rates of any surgical implant with 50% failing in the first two years after placement[4]. The most common symptom of shunt failure is a headache and patients routinely present to hospital with headache suspecting shunt failure. This in turn leads to hospital admission, neurological consultations and investigations including brain imaging. However, two-thirds of these admissions are false alarms[5]. A large portion of hydrocephalus occurs from birth, and its management is particularly difficult in non-verbal children. The unmet clinical need is for a system that allows for the accurate long-term measurement of ICP at home, enabling patients and physicians to make data driven treatment choices rather than relying on nonspecific symptoms.

The implant is designed to be inserted directly into the cerebral cortex during the same neurosurgical procedure to insert a shunt for CSF drainage. The dimensions of the sensor are 2.0 × 3.6 × 20.1 mm, similar to the diameter of a CSF shunt (Fig. 1B). The implant weighs

0.28 g and has a density of 2.0 g/cm³ with all tissue contacting materials comprised of borosilicate glass. This density is significantly less than devices using titanium or ceramic casing although more than brain tissue (1.04 g/cm³)[6].

Long-term implantation of pressure sensing devices has achieved regulatory approval for cardiac applications but a change in the measured signal over time (drift) remains a concern for regions of the body where recalibration is not feasible such as in the brain[7]. We have taken a novel development path where the implant transduces pressure into a digital signal and then reports all necessary information for the measurement to the reader digitally, with no external calibration values required. This method eliminates inaccuracy due to environmental interference and enables a wide range of use conditions encountered with at-home monitoring.

The pressure-sensing implant uses a hermetic capacitive transducer, with a separate cavity for electronics (Fig. 1C) to achieve accuracy that exceeds standards (NS28) for ICP monitoring over the short (Fig. 2A) and long-term, as evidenced by studies in which real-time drift was <2 mmHg over a one-year period (Fig. 2B). Accelerated aging indicated that total drift is expected to be <5 mmHg over a 10-year period (Fig. 2C). Both results demonstrated exceptional stability, and no failures nor data transmission issues were observed. Commercial wafer-scale glass processing enables both the scalable manufacturing and uniform quality, which are essential for a medical device.

The electronics within the sensor are limited to a hermetic cavity sized 16.0 × 2.5 × 1.1 mm. The implant uses resonant wireless power transfer from an external wand, avoiding the need for an implanted battery with its attendant size and longevity issues. The small implant

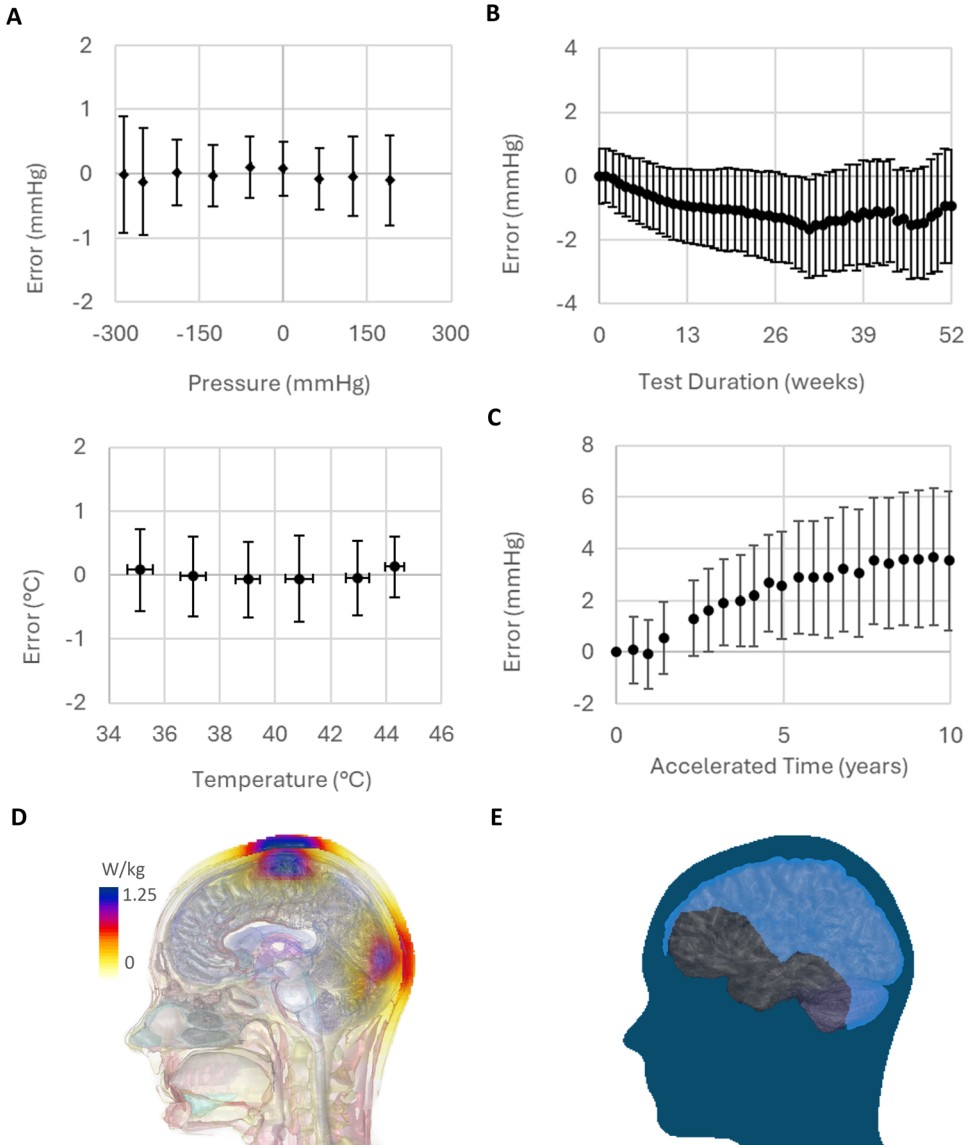

**Fig. 2 | Performance and Safety Characterization of the Implantable ICP Sensor System. A** Sensor pressure and temperature accuracy over the full operational gauge pressure and temperature ranges (nominal barometric of 760 mmHg, $N = 462$ measurements, mean ± SD) as per ICP monitoring standards (AAMI NS28). **B** Real-time zero-point drift testing showing that sensor drift is acceptably small for clinical use ($N = 16$ implants, mean ± SD). **C** Accelerated drift testing using 85 °C incubation ($N = 15$ implants, mean ± SD, acceleration factor = 23.4). **D** Worst case peak spatial average SAR (per IEEE C95.1) lower than the 2 W/kg safety limit. **E** The light blue overlay shows the areas in the brain where a measurement can be made with the implant in the worst-case orientation; if implant orientation is ideal any place within the brain becomes accessible.

size and deep implantation present challenges to safe power transfer without causing excessive tissue heating. Operating at low MHz frequencies, where specific-absorption-rate (SAR) is low, minimizes tissue heating (Fig. 2D). Wireless power makes data transfer challenging using a miniaturized implant due to strong interference from the wireless power field. We developed a new data transfer method where power and data are transferred sequentially. Resonant-tank-clamping phase-shift-keying (RPSK), allows efficient data transfer up to 12 cm between the implant and a reader coil (Fig. 2E). The design enabled sampling of the pressure at 50 Hz and was therefore sufficient to resolve the cardiac-related frequency in ICP waveforms. ICP data was received by a handheld wand which connected to an App running on the patient's phone and then to a cloud-based portal for remote clinical review. (Fig. 1A). Although the current implant measures pressure, the basic device is capable of other sensing modes or stimulation with the addition of surface electrodes.

The implant was found to be biocompatible with excellent hermeticity (exceeding MIL-STD-883L) and well-suited for long-term implantation. Histological examination of cortical tissue from sheep implanted with sensors for up to 12 months revealed a stable tissue response with minimal gliotic rim (thickness <100 μm, Fig. 3A). There was no evidence of neurotoxicity, changes in animal behaviour or widespread systemic effects (e.g. hematology, blood chemistry). Importantly, the implant did not migrate in the cortical tissue and stayed within the measurement detection limit of 2.4 mm as evidenced by serial radiographs. Histological comparison to a shunt tube placed in adjacent cortical tissue indicated a similar or smaller tissue response.

Approval for a first-in-human trial was obtained from the New Zealand Health and Disability Ethics Committee (ClinicalTrial.gov NCT06402786). Twenty patients (12 male, 8 female, 10 adult and 10 children; age range 18 months to 84 years) undergoing first shunt insertion or revision surgery had the ICP sensor implanted during the

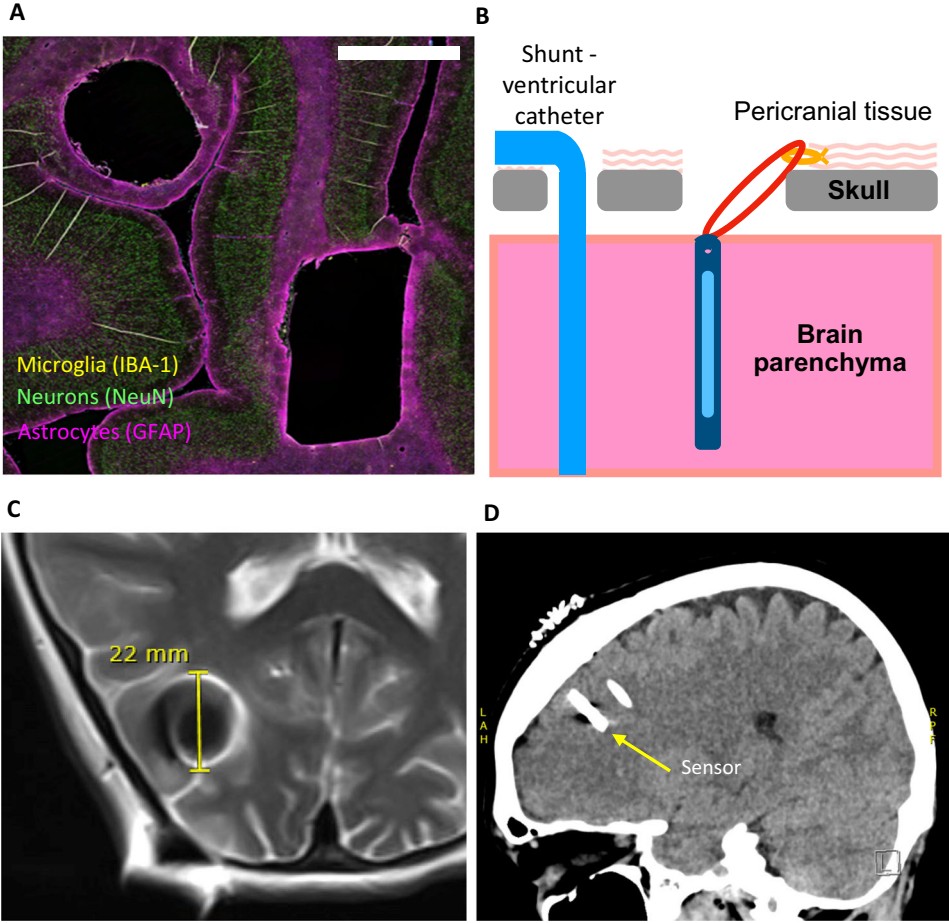

**Fig. 3 | Long-Term Tissue Response, Implant Positioning, and Post-Operative Imaging of the ICP Sensor. A** Immunohistochemistry of sheep brain tissue of the round shunt tubing adjacent to the sensor at 6 months post implantation (scale bar 2.5 mm) (6 sheep had histology performed). Green staining shows neurons stained with NeuN and indicates no widespread neuronal loss. Yellow shows quiescent microglia (IBA-1), indicating stabilisation of the tissue response at 6 months post-implantation. Pink shows astrocytes (GFAP) indicating minimal glial encapsulation. **B** Schematic showing the placement of the sensor directly within the parenchyma via a separate burr hole from the shunt. **C** Post-operative MRI image illustrating an approximate 22 mm artifact associated with the sensor. **D** Post-operative CT image showing the sensor adjacent to a shunt catheter within the cortex.

same operation, adding an average of 6 min of extra surgical time (informed consent was obtained from all subjects or, in the case of children, from caregivers). Participants included individuals with normal pressure hydrocephalus (NPH), as well as other congenital and acquired forms of hydrocephalus. The sensor was inserted via a separate burr hole (diameter ~9 mm and 1–2 cm away from the shunt burr hole) into the cerebral parenchyma, parallel to the ventricular catheter, allowing the top of the sensor to sit flush with the cortical surface. The separate burr hole does present additional risk of bleeding but is considered a standard neurosurgical procedure and no complications were noted. A short length of suture material was passed through a hole in the sensor and tied to pericranium tissue. This served as a locator thread rather than a tether in case the implant was needed to be removed at a later date (Fig. 3B). Patients received standard pre- and postoperative neurosurgical care. After discharge, participants were instructed to measure their ICP at home daily for the first two weeks, and then every second day for the following three months. The longest monitoring period is currently 600 days and ongoing. Post-procedure CT scans at 3 months were used to determine if devices showed signs of movement. In 19 patients there was no movement of sensors exceeding the measurement error of the Stealth analysis system (Medtronic) (~2 mm) but in one patient, a 5 mm movement of the sensor inwards was observed. There were no signs of oedema around the sensors.

(Fig. 3D). The implant produced a small, ~22 mm, distortion field under 3 T MRI imaging. (Fig. 3C).

There were no adverse events related to the sensor, or its use at the 3-month primary safety endpoint, and no sensor failures occurred with over 2500 participant recorded ICP measurements to date. Each clinical event such as shunt failure or hospitalization was assessed by an independent safety monitoring committee as to the possibility that the event was device related. None were noted. The median ICP values ranged from ~20 to 27 mmHg depending on body position and hydrocephalus aetiology (Fig. 4A). The median amplitude of the cardiac related pulse wave of ICP in asymptomatic patients was generally between 1 and 3 mmHg across the study period (Fig. 4B). If a large amount of encapsulation of the implant had occurred, we would expect to have seen a reduction in the amplitude of this waveform i.e. loss of signal fidelity and dampening of the frequency components of the ICP signal. The cardiac-related pulse wave amplitude was not affected by body position. In patients who remained clinically stable with no concerning symptoms or hospital admissions, mean ICP remained stable over extended periods of time and postural differences could easily be distinguished (Fig. 4A). This was interpreted as a clinical confirmation of the system's accuracy and stability, and appropriate for ICP monitoring requirements.

Evidence of clinical utility was confirmed, specifically a pediatric patient returned to hospital 74 days following implantation with a

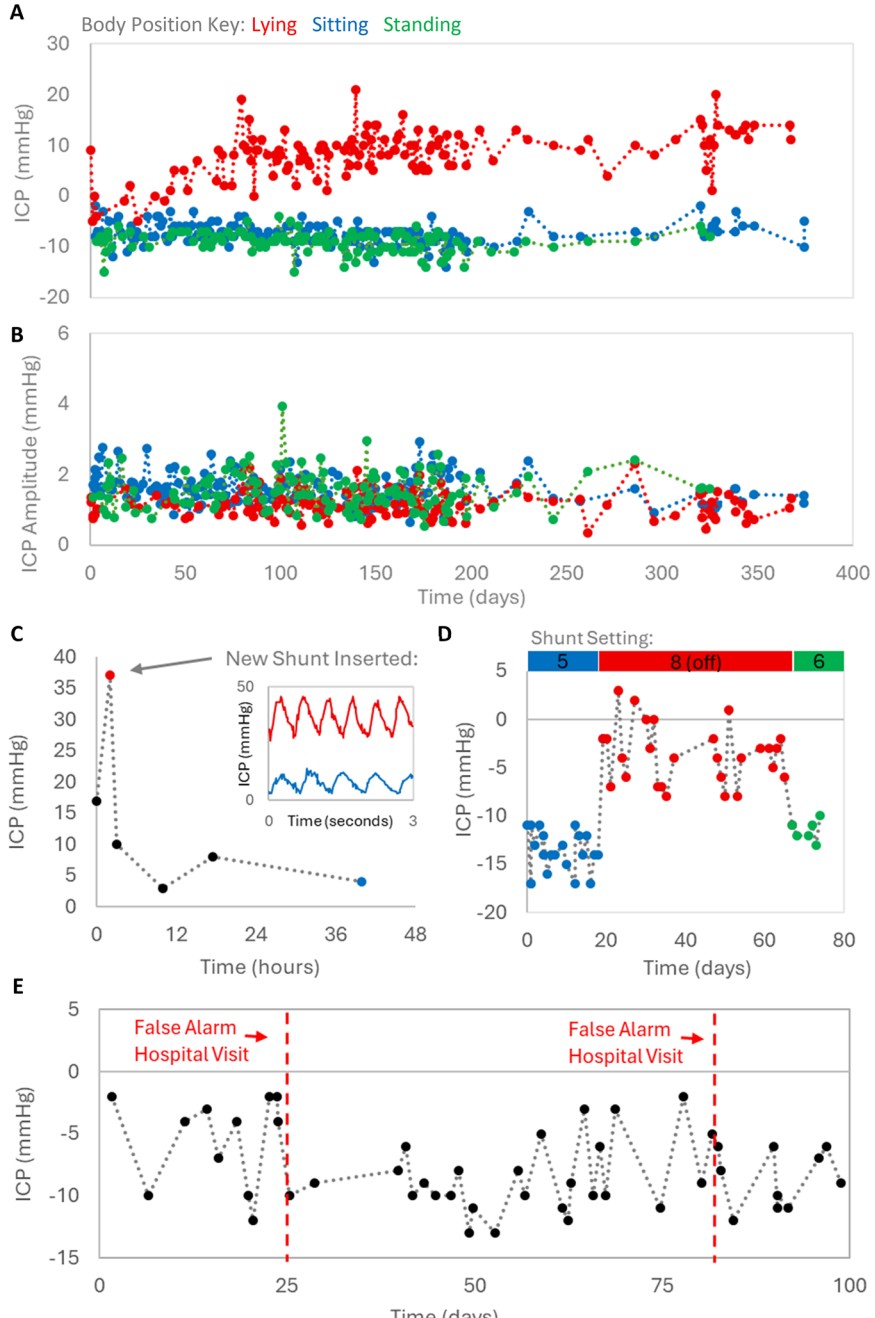

**Fig. 4 | Real world ICP Monitoring: Long-Term Stability, Shunt Responses, and Clinical Case Examples. A** ICP data from one adult patient over a 1-year period from the time of shunt and sensor insertion under different postures, all data were obtained in the home environment and illustrate the stability in an asymptomatic patient. **B** The amplitude of the cardiac-related pulse wave from the same adult patient over a 1-year period. **C** ICP values obtained from a 19-month-old child pre- and post-shunt revision with the amplitude of the ICP waveforms in the insert. **D** ICP values from a 78-year-old patient suffering from normal pressure hydrocephalus who had shunt valve adjustment. **E** ICP values obtained from a 7-yr old over 3 months where the parents took the child to hospital twice with a suspected shunt failure but subsequently testing positive for influenza i.e. false alarm.

severe headache. ICP values were found to be significantly elevated. A CT scan showed dilated ventricles, and the ventricular catheter was confirmed to be blocked at subsequent surgery. Following shunt revision, the ICP returned to normal values accompanied by resolution of headache (Fig. 4C). In one adult with NPH, the shunt valve was temporarily switched to the 'virtual off' position due to concerns of over-drainage. This resulted in a relapse of the patient's NPH-related symptoms of gait disturbance and a marked increase in both the mean ICP and waveform amplitude from 2–3 mmHg to 6–8 mmHg. After subsequently re-opening of the valve, both the mean ICP and pulse amplitude decreased accordingly (and gait disturbance resolved)

(Fig. 4D). A 7-year-old boy was taken to hospital with ear pain 25 days after shunt revision and sensor placement surgery. A CT brain scan showed no significant changes. He subsequently tested positive for Influenza B, and the episode was deemed a false alarm and not a shunt failure. ICP recorded by the system remained within his normal range throughout. Subsequently at 82 days post-surgery he was again taken to a regional hospital with fever, unsteadiness and acute confusion. Following an overnight hospital stay, the patient was transferred to a neurosurgical unit for observation. A CT scan revealed no new intra-cranial findings and he was discharged home. This proved to be another false alarm, with the ICP recorded by the sensor remaining

within his normal range throughout, and no subsequent shunt revisions have been required to date (Fig. 4E). The interpretation of these results is that the use of ICP monitoring in a home environment could have avoided these hospitalisations. Across the cohort, all patients and caregivers surveyed at three months reported a significant reduction in anxiety associated with their condition as a result of ICP measurement.

The Miethke Sensor Reservoir (M.Scio) provides an alternative method for measuring intracranial pressure (ICP) via telemetry[8]; however, its design differs substantially from the system described here. The M.Scio sensor is integrated with the shunt system and therefore measures pressure within the shunt pathway (the system can also be used as a stand-alone monitoring device with the pressure sensor connected to the ventricular catheter). As a result, accurate ICP measurement depends on an unobstructed transmission of cerebrospinal fluid and pressure to the sensor. Because a substantial proportion of shunt malfunctions arise from blockages in the ventricular catheter[9], an ICP value may still be obtained even when it no longer reflects true intracranial pressure. In contrast, our sensor is independent of shunt type or patency and measures ICP directly within the brain parenchyma. A second major distinction is that the M.Scio system is intended for use by physicians within a hospital setting and is not designed for patient-operated measurements. Hydrocephalus patients place a substantial burden on neurosurgical services, yet the majority of hospital presentations for suspected shunt malfunction do not result in surgical intervention and are ultimately deemed false alarms[5]. Home-based ICP measurement has the potential to provide physicians with timely, remote access to physiological data, enabling earlier decision-making while avoiding unnecessary hospital visits and reducing strain on healthcare resources. A final significant difference concerns data handling: the M.Scio system records measurements onto an SD card inserted into the reader, requiring physician involvement for data retrieval and offline analysis. In contrast, our system has been designed for seamless remote monitoring. ICP measurements are transmitted from the wand via Bluetooth to a smartphone application and subsequently to a cloud-based clinical portal, enabling immediate access and streamlined review.

Clinical interest in pressure sensing has increased over the past decade with the concept of using the pressure readings from the pulmonary artery as an input to titrate drug therapy in heart failure. The Cardiomems device (Abbott Laboratories) might be considered the first generation, the Endotronix device from (Edwards Life Sciences) the second generation and now our fully digital pressure sensor with novel advances in microfabrication to reduce drift and improve fidelity of pressure sensing in general as the next generation. Although not specifically tested in the present study, the implant is small enough to be placed in other regions of the body to measure pressure.

The major advance of the developed technology is the demonstration of clinical safety, accuracy and reliability of remote ICP measurement using a discrete micro-implant within the human brain over an extended period in both adults and children. Children are recognized to have the worst outcomes in terms of hydrocephalus management due to repeated shunt failures[5]. The additional paucity of novel medical devices developed with children in mind means paediatric hydrocephalus remains an area of profound unmet clinical need, where innovation lags and children bear the consequences of systemic underinvestment in tailored solutions. The ability to easily record ICP in a home environment is a significant advancement in the management of hydrocephalus. The extremely high failure rate of CSF shunts coupled with the high incidence of false alarms, generates substantial work for hospitals as well as stress and inconvenience for patients and their families. A technology that offers real-time physiological data remotely heralds a significant advance in the care paradigm for these patients.

## Methods
### Ethics
Every experiment involving animals, or human participants carried out following a protocol approved by an ethical committee. In the case of human participants, each participant gave informed written consent.

**Methods used for power calculations.** Implantation power transfer range limits the depth at which the implant can operate. The depth is orientation dependent, with the range being longer when the implant coil is parallel (perfectly aligned) with the wand coil. However, perfect orientation of the implant with the wand coil may not be possible and here we determine what the worst-case power transfer range is. This assumes the implant can be in any orientation, though many of these scenarios are unlikely, as a conservative means to understand range limitations.

Moving the wand changes the magnetic field strength and orientation, and this can be used to "find" the implant where the wand to implant orientation and distance between them allows sufficient power transfer. In bench top testing it was found that the implant needs 28 $\mu T_{RMS}$ at minimum to power when ideally oriented. This is found by taking the dot product of the magnetic field vector (B) and the orientation vector orthogonal to the implant coil plane (a). We developed a numeric model to find the region inside the head where the implant can always receive power, regardless of orientation. The MIDA head model was used as the base for our model; it is a high resolution model of the adult head and neck that was created by the FDA, Center for Devices and Radiological Health, and the IT'IS Foundation and is often used in modelling studies[10]. The head model was simplified to a mesh of the outside of the head only using a combination of Laplacian smoothing and Quadric Edge Collapse Decimation in the open-source software Meshlab (version 2023.12). The interior of the head mesh was discretised as a rectilinear grid with a resolution of 1 mm in x, y and z. Feasible positions for the wand on the head were found with a non-linear least squares optimisation search routine that minimises the distance from the wand coil to the head from an initial position. In total, 720 wand positions around the head mesh were simulated. For each position, the B-field vectors were calculated at every grid point inside the head mesh (~8 million points). We recorded the lowest B.a value (available magnetic field) at each point and plotted it across the sagittal plane of the MIDA model. Areas in light blue on Fig. 2E are locations where the implant can be powered, and pressure recorded, regardless of the implant orientation. Areas in dark blue on the same figure show locations where the implant can be powered only with a suitable orientation.

**SAR and power transfer.** The system uses magnetic induction to transfer power to the implant. Specific Absorption Rate (SAR) is a measure of the power deposited in tissue from circulating current arising from exposure to electromagnetic fields from the wand. It quantifies heating in W/kg and is safe below 2 W/kg averaged over any 10 g (PSAR- Peak spatial average SAR) per safety standard IEEE C95.1; these limit local heating effects in areas where the magnetic field is high. There are also whole-body SAR limits in IEEE C95.1 (0.08 W/kg) which prevent whole body temperature rise becoming unsafe.

The worst-case SAR generated by the wand was determined in Sim4Life using 3 models: Mida (Adult), Athena (3.5 years) and Martin (29 months). The wand was modelled as a 15 cm diameter circular current carrying coil using the quasistatic low frequency solver and 10 g averaged peak and whole-body SAR were extracted (peak spatial specific absorption rate, psSAR). Wand coil current was

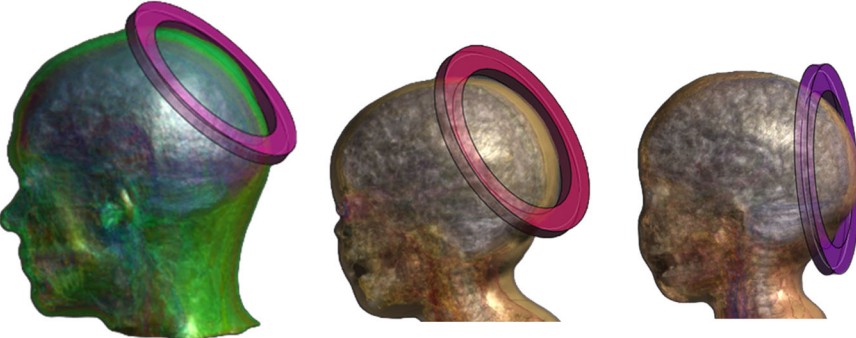

**Fig. 5 | Worst case orientations for psSAR identified in the Mida (adult), Athena and Martin (paediatric) models (left to right).**

**Table 1 | Maximum SAR values and total absorbed power from the whole-body exposure requirement using 3 models**

|  | Limit | Mida (ADULT) | Martin (3.5 years) | Athena (29 months) |
|---|---|---|---|---|
| Total power absorbed |  | 1.63 W | 1.14 W | 1.28 W |
| PSAR (continuous) |  | 3.61 W/kg | 3.59 W/kg | 3.67 W/kg |
| PSAR averaged over 6 minutes | 2 W/kg | 1.20 W/kg | 1.20 W/kg | 1.22 W/kg |

11 $AT_{RMS}$ (AT = Amp Turns) which is the maximum the wand can generate. The location of the coil which causes the highest SAR varies with anatomy and is typically maximised for positions where the whole coil is as close as physically possible to the head. Maximum values were found by moving the loop around each model and recording the SAR for each location, which is shown for each of the models in Fig. 5.

Table 1 shows the identified maximum SAR values, total absorbed power resulting from the whole-body exposure requirement from Sim4Life using three models. IEEE C95.1 specifies that PSAR is averaged over 6 minutes and total absorbed power is averaged over 30 minutes when calculating whole body SAR, and the averaged values are given in each case for a 120 second measurement. This is the maximum allowed by the wand, with a typical measurement taking 10 to 15 seconds. These results show that the wand complies with the standard for adults and children for peak spatial averaged SAR. Total absorbed power suggests weights down to approximately 1 kg are safe with a measurement time limit of two minutes.

**Wireless power and data**. A sequential data transfer method called resonant-tank-clamping phase-shift-keying (RPSK) was developed to enable wireless power and data transfer to implantation depths of up to 10 cm. This method separates power and data transfer allowing small signals from the implant to be detected without interference from the strong wireless power transfer field. To implement and test this method, an application-specific-integrated-circuit (ASIC) was designed and fabricated in 180 nm CMOS. The ASIC includes a rectifier and power management circuitry to convert the received wireless power into a DC supply, an oscillator driver and phase shifting switch to drive oscillations and phase shifts in the implant wireless power inductor and capacitor (LC) tank for data transfer and control circuitry to switch between power and data transfer when the wireless power field is turned off. During data transfer, energy stored on a capacitor bank within the implant is used to power the ASIC. Oscillations are driven on the wireless power LC tank. Relatively little power is required as the LC tank has a high quality factor with low resistance. Phase shifts are introduced by clamping the current in the LC tank at maximum current and minimum voltage for half of the period of the oscillating voltage. The short time of the clamping combined with the low resistance of the clamping switch ensures most of the energy in the resonant tank is conserved at the new phase. Data transfer range was tested using a 6 mm dimeter coil at the implant and a 50 mm coil to receive the data transfer. A RF receiver was developed using a Costas loop to demodulate PSK data transfer. Data was transferred over a range of 150 mm with a measured bit error rate of $3.8 \times 10^{-8}$.

**Sensor accuracy test**. The initial accuracy of the sensors was evaluated using a custom test rig that measured a standard reading from the implant (Implant ID 1607, 504 samples or approximately 10 s of data at 50 Hz, acquired over no more than 30 s) with a basic wireless reader in controlled environmental conditions to determine sensor accuracy across a relevant range of physiologic conditions. The condition ranges tested were from 475 to 950 mmHg (absolute) at temperatures from 35 to 45 °C. During the test, equilibrium pressure and temperature steps were maintained to ±0.5 mmHg and ±0.5 °C respectively. All readings were taken with the implant under 1 cm pH 7.4 Dulbecco's Phosphate Buffered Saline (dPBS, Gibco 21-600). Pressure error was calculated by subtracting the implant absolute pressure from an adjacent calibrated barometer (Mensor CPT6100) and a fixed dPBS head pressure (1 cm ≡ 0.73559 mmHg). Data presented was from a single sensor and is typical of the sensor performance. Sensor accuracy testing was conducted on each sensor prior to sterilisation, with acceptance being that total error across the range was within ±2 mmHg and ±2 °C, a limit derived from ANSI/AAMI NS28:1988 (R2015).

**Real time zero-point drift test**. The real time drift test measured implant pressure error in a simulated implanted environment. A custom "drift rig" enabled automated wireless data collection from multiple implants maintained in test conditions designed to mimic conditions experienced during the implant's intended use as an implanted sensor for ICP monitoring. In summary, implants under test were kept at 38 °C in pH 7.4 Dulbecco's Phosphate Buffered Saline (dPBS, Gibco 21-600) in containers vented to atmosphere. Daily readings, consisting of a time course of pressure data, was read from each implant using wireless power settings that mimicked the ICP monitor use case (504 samples or approximately 10 s of data at 50 Hz, acquired over no more than 30 s). Zero-point readings were then calculated by subtracting the implant absolute pressure readings from an adjacent calibrated barometer (Mensor CPT6100) and a fixed dPBS head pressure (1 cm ≡ 0.73559 mmHg). Test automation and data logging was performed with a computer running a custom control program (NI LabVIEW). The dPBS level and other test conditions were checked periodically to ensure system stability. For the data presented, measurements were taken over a 52-week period and readings were converted to zero-point by subtracting the t = 0 value.

**Accelerated drift test.** Sample preparation and data acquisition procedures for the accelerated lifetime test were identical to the zero-point drift test described above. The difference for accelerated testing is that the implants were held at elevated temperature. Specifically, implants were kept at 85 °C between reading sessions; during a reading session the implants were cooled to 39 °C, readings were then taken at 39 °C, finally the implants were returned to 85 °C. Over the weekly cycle, implants were at 85 °C for 96% of the time (162/168 hours). Data is presented using the accelerated time scale. An acceleration factor of 23.3 was calculated using standardised Arrhenius methods (ASTM F1980 – 16 "Accelerated Aging of Sterile Barrier Systems for Medical Devices") with a working temperature of 39 °C, the elevated temperature of 85 °C, a reaction rate coefficient of 2 and a duty factor of 96% (162/168). This calculation is conservative in that all implant test time spent under the elevated temperature was at the working temperature. For the data presented, weekly measurements were taken over a 22-week period, representing approximately 10 years of accelerated time. As above, readings were converted to zero-point by subtracting the $t = 0$ value

**Histology methods.** The detailed methodology for histology has previously been described[11]. Briefly, following tissue processing, tissue blocks containing the region where the sensor and shunt were inserted, as well as a control from an unaffected cortical region from the same sheep were cut in a transverse orientation, enabling the cross-section of the implant and shunt to be analysed. 10 μm thick sections were subject to heat-induced epitope retrieval with 10 mM sodium citrate buffer, pH 6.0 (GFAP and collagen IV) or citric acid buffer, pH 6.0 (IBA-1), blocking, and antibody incubations. Rabbit polyclonal, primary antibodies used were: glial fibrillary acidic protein (GFAP), 1:1000 (Dako, Z0334), ionized calcium-binding adapter protein (IBA-1), 1:500 (Wako 019-19741), and collagen IV (Col4A), 1:300 (Biorbyt 340147). Goat anti-rabbit IgG (H + L) Alexa Fluor 647 was used as the secondary antibody for single-label fluorescent immunohistochemistry and Hoechst 33342 (Molecular Probes, H1399), 1:10,000, as a nuclear counterstain.

**Surgical methods.** During surgery to place, or replace, the shunt a second burr hole was drilled in the skull near the burr hole for the ventricular catheter and using the same scalp incision. The mean burr hole size for the implant placement was $9.3 \pm 0.3$ mm (range 7 to 11 mm). The dura and pia membranes were excised and the implant inserted into the cortex perpendicular to the skull until the top of the implant was approximately level will the surface of the cortex. The locator thread was secured to pericranial tissue outside the burr hole and a small piece of gel foam was placed in the burr hole over the implant. The additional surgery time required to drill the extra burr hole and insert the implant was $6.7 \pm 0.8$ minutes (range 3 to 12 minutes).

### Reporting summary
Further information on research design is available in the Nature Portfolio Reporting Summary linked to this article.

## Data availability
All data supporting the findings of this study are available within the article and its supplementary files. Any additional requests for information can be directed to, and will be fulfilled by, the corresponding authors. Source data are provided with this paper and available at https://figshare.com/s/06d81cd5988d1742d1bd. Source data are provided with this paper.

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

## Acknowledgments
Toby Jackson of the Auckland Bioengineering Institute kindly assisted by adapting the MIDA head models to a surface appropriate for the minimum-power modelling work. Funding for this work was provided by the Health Research Council of New Zealand, the Ministry for Business Innovation and Employment (Endeavour Fund), Cure Kids and Kitea Health Ltd.

## Author contributions
S.C.M. conceived the project, designed the study framework, secured funding and wrote the initial manuscript draft. B.E.W. developed the overall implant design, developed bench test plan and analysed data. S.G. developed the preclinical experimental methodology, performed the animal surgeries, collected physiological data and managed the clinical trial. P.H. performed neurosurgery on patients. R.J.G. developed the implant communications scheme and internal electronics. D.P.L. developed bench test plan and analysed data. S.K. performed the animal surgeries and analysed preclinical and clinical physiological data. Q.B. performed analysis of S.A.R. and power calculations. S.T. performed histological analysis and interpreted tissue-response findings. M.K. developed firmware for the overall system and the patient app. DJM managed the clinical trial. J.A.W. provided supervision to S.K. and PAB reviewed clinical data as part of monitoring for adverse events and overall operation of the clinical trial. D.M. developed the wireless power transfer system and external electronics. All authors reviewed, edited, and approved the final manuscript. S.C.M., B.E.W. and S.G. contributed equally.

## Competing interests
Authors Malpas, Wright, Guild, Gallichan, Leung, Kondo and McCormick are employees and shareholders in the company Kitea Health Ltd. The authors declared no other competing interests.
