## [Transparent Peer Review file · Nature Communications]

Long-term brain pressure monitoring via a discrete microimplant; a first-in-human safety and initial efficacy trial in adults and children with hydrocephalus

Corresponding Author: Professor Simon Malpas

Version 1:

Reviewer comments:

Reviewer #1

(Remarks to the Author)

This safety and tolerability study of the Kitea ICP system is an excellent work in its field. The developers of the system have succeeded in elaborating an implant similar to the diameter of a ventricular catheter, with low weight and density, no drift and enabling sampling of pressure at 50 Hz.

The clinical need of remote telemetric ICP measurements in patients with shunted hydrocephalus is real and not fully met worldwide. Over time there has been several devices trying to do the same (more or less) as the Kitea ICP system¹. There have been challenges with prior systems and the only commercially available telemetric ICP measuring system on the market now, the Miethke Sensor Reservoir (M. Scio) was introduced in 2015². This is commented in the Clinical Trial protocol but not in the article, plausible giving the reader the perception that the Kitea ICP system is ground braking in a field where a clinical option already exists. What are the advantages of the Kitea ICP System to the M. Scio?

The authors state that there is an unmet clinical need for a system that allows for the accurate long-term measurements of ICP at home, enabling patients and physicians to make data driven treatment choices rather than relying on nonspecific symptoms. In my perspective the largest advantages of the telemetric ICP measurements are for the physician in the clinic. If the patient has symptoms that could imply shunt failure, they should seek medical care and be examined by a physician inclusive ICP measurement. To make a measurement at home could be sufficient with low grade symptoms (e.g. headache) but for example a non-verbal child with nonspecific symptoms would still need examination of a physician. The authors report no adverse events at 3-months follow-up, does that include per- and postoperative complications?

The methodology is sound, and the work is impressive in several ways. The Clinical Trial Protocol is well thought out and very informative. The developers of the Kitea ICP system have outlined the problems with ICP measurements and managed to overcome these in an impressive way.

The work supports the conclusion that the developed technology demonstrated clinical safety, accuracy and reliability of remote ICP measurements. I suggest this work to be published with addition of a description of existing telemetric ICP systems and if there is any advantage with the Kitea ICP system.

1. Yangi, K. et al. Telemetric intracranial pressure monitoring in patients with hydrocephalus: a systematic literature review. *Front. Pediatr.* 13, 1632216 (2025).

2. Ertl, P., Hermann, E. J., Heissler, H. E. & Krauss, J. K. Telemetric Intracranial Pressure Recording via a Shunt System Integrated Sensor: A Safety and Feasibility Study. *J. Neurol. Surg. Part Cent. Eur. Neurosurg.* 78, 572–575 (2017).

Sara Magnéli, Consultant Neurosurgeon, Uppsala University Hospital, Sweden

Reviewer #2

(Remarks to the Author)

The authors report successful design, fabrication, regulatory approval, and initial placement of pressure monitoring devices into the brains of adult and pediatric patients with hydrocephalus, allowing noninvasive estimation of intracranial pressure and even waveform of the pressure transduction. This is a noteworthy result.

Such a device could significantly improve and make more quantitative the management of possible shunt malfunction and other intracranial pressure-related issues in patients with hydrocephalus. This is not the first device to allow noninvasive measurement of ICP (see, for example, the Radionics Tele-Sensor), but it has the advantage of sensing intraparenchymal pressure, rather than intraluminal shunt pressure within the shunt (which may not adequately reflect true intracranial pressure if there is an obstruction).

The paper gives a strong argument that even "home" measurements of intracranial pressure (ICP) with this device would be potentially useful. However, the illustrations of clinical support for the value of the measurements are somewhat anecdotal. Descriptions are provided of "false alarm" shunt events versus true malfunctions, but even this dichotomy oversimplifies the actual state of shunt management by neurosurgeons. This is probably reasonable in a "proof of concept" presentation, but evaluation of the true value of the measurements to clinicians would depend on a more detailed, presumably Bayesian analysis--potentially beyond the scope of this paper. In the introduction to the main section, there seems to be an implication that this device could resolve issues with diagnosing shunt problems in non-verbal children. I would be very hopeful that this is the case--but evaluation of this question is more nuanced than the clinical histories in the results of the paper.

More minor comments:

The very low mass of the device is impressive, but the potential risks and problems with the device may depend more on the need for a second penetration of the brain (potential bleeding) and the MRI artifact (loss of diagnostic precision). I found the multiple references to the low mass a bit irrelevant compared to these actual clinical concerns.

The device appears similar in functionality to the CardioMEMS device (Abbott Laboratories) intended for a different location of pressure measurement (pulmonary vasculature). It appears that this device has a distinct construction, different materials, etc.--yet the claim is made that this device could be used anywhere in the body for pressure measurement (and, potentially, other things.) For a brief announcement of a new technological advance as in this paper, it would be helpful to distinguish from other similar devices available.

I am happy to hear of these results and look forward to availability of the device. The authors should be congratulated for this work!

Joseph R. Madsen, MD

Version 2:

Reviewer comments:

Reviewer #1

(Remarks to the Author)

Dear authors,

Thank you for your review of the manuscript. There are only two minor details in the added section of Miethke Sensor Reservoir that need to be addressed. Firstly, the M.Scio sensor is not integrated within the shunt valve, it is connected to the ventricular catheter proximal and to a catheter distal (which is connected to the shunt valve). It could also be used as a stand-alone monitoring device only connected to a ventricular catheter without a shunt. Secondly, because of this it is possible to connect the M. Scio to any type of shunt valve via a distal catheter, not only Miethke shunt valves.

Best regards,

Sara Magnéli, MD, Uppsala University Hospital

Reviewer #2

(Remarks to the Author)

I am satisfied with the responses of the authors to my questions.

REVIEWER COMMENTS

Reviewer #1 (Remarks to the Author): Our responses are in red

This safety and tolerability study of the Kitea ICP system is an excellent work in its field. The developers of the system have succeeded in elaborating an implant similar to the diameter of a ventricular catheter, with low weight and density, no drift and enabling sampling of pressure at 50 Hz.

The clinical need of remote telemetric ICP measurements in patients with shunted hydrocephalus is real and not fully met worldwide. Over time there has been several devices trying to do the same (more or less) as the Kitea ICP system. There have been challenges with prior systems and the only commercially available telemetric ICP measuring system on the market now, the Miethke Sensor Reservoir (M. Scio) was introduced in 2015. This is commented in the Clinical Trial protocol but not in the article, plausible giving the reader the perception that the Kitea ICP system is ground breaking in a field where a clinical option already exists. What are the advantages of the Kitea ICP System to the M. Scio?

Author response:

Thank you for the positive comments. As noted, “the clinical need of remote telemetric ICP measurements in patients with shunted hydrocephalus is real and not fully met worldwide”. We wish to clarify why the presented technology is ground breaking over and above that previously developed e.g. M.Scio.

- The engineering of a complete medical device acting as a pressure sensor in a package 0.28 g in weight
- The placement of the discrete sensor within the cortex and demonstration of safety and efficacy in children and adults
- The ability to measure ICP in the home environment i.e. without physician involvement to potentially reduce the burden on patients, families and physicians in managing these complex patients.

The reviewer is correct that the article does not cover the alternative telemetry system Mscio. In the revised manuscript we now include the paragraph: “The Miethke Sensor Reservoir (M.Scio) provides an alternative method for measuring intracranial pressure (ICP) via telemetry; however, its design differs substantially from the system described here. The M.Scio sensor is integrated within the shunt valve and therefore measures pressure only within the shunt pathway. As a result, accurate ICP measurement

depends on an unobstructed transmission of cerebrospinal fluid and pressure to the sensor. Because a substantial proportion of shunt malfunctions arise from blockages in the ventricular catheter⁹, an ICP value may still be obtained even when it no longer reflects true intracranial pressure. In contrast, our sensor is independent of shunt type or patency and measures ICP directly within the brain parenchyma. A second major distinction is that the M.Scio system is intended for use by physicians within a hospital setting and is not designed for patient-operated measurements. Hydrocephalus patients place a substantial burden on neurosurgical services, yet the majority of hospital presentations for suspected shunt malfunction do not result in surgical intervention and are ultimately deemed false alarms⁵. Home-based ICP measurement has the potential to provide physicians with timely, remote access to physiological data, enabling earlier decision-making while avoiding unnecessary hospital visits and reducing strain on healthcare resources. A final significant difference concerns data handling: the M.Scio system records measurements onto an SD card inserted into the reader, requiring physician involvement for data retrieval and offline analysis. In contrast, our system has been designed for seamless remote monitoring. ICP measurements are transmitted from the wand via Bluetooth to a smartphone application and subsequently to a cloud-based clinical portal, enabling immediate access and streamlined review.

The authors state that there is an unmet clinical need for a system that allows for the accurate long-term measurements of ICP at home, enabling patients and physicians to make data driven treatment choices rather than relying on nonspecific symptoms. In my perspective the largest advantages of the telemetric ICP measurements are for the physician in the clinic. If the patient has symptoms that could imply shunt failure, they should seek medical care and be examined by a physician inclusive ICP measurement. To make a measurement at home could be sufficient with low grade symptoms (e.g. headache) but for example a non-verbal child with nonspecific symptoms would still need examination of a physician.

Author response:

Our system is not designed to replace the need for physician review but rather enabling information for the decision to be made earlier or remotely from the physician. We are aware of many patients living remotely from a neurosurgical level hospital and thus families experience high levels of anxiety associated with urgent travel and assessment at hospitals. Sometimes transfers from regional hospitals to main centers need to be via air ambulance. We suggest that having a simple ability to measure ICP in the home environment and that data made available to physicians has the ability to substantially change the care paradigm for these complex patients. We we hope this clarifies our intent and don't believe there is any need to change the original text.

The authors report no adverse events at 3-months follow-up, does that include per- and postoperative complications?

Author response:

The specific terminology used in the manuscript and clinicaltrial.gov is around the safety endpoint “that there we no adverse events attributable to the device”. This language is traditional for device trials. Each clinical event such as shunt failure, hospitalization was assessed by an independent safety monitoring committee as to the possibility that the event was device related. None were noted. We have amended the manuscript to clarify this point.

The methodology is sound, and the work is impressive in several ways. The Clinical Trial Protocol is well thought out and very informative. The developers of the Kitea ICP system have outlined the problems with ICP measurements and managed to overcome these in an impressive way.

The work supports the conclusion that the developed technology demonstrated clinical safety, accuracy and reliability of remote ICP measurements. I suggest this work to be published with addition of a description of existing telemetric ICP systems and if there is any advantage with the Kitea ICP system.

Author response:

We thank the referee for their thorough review, and the suggested changes have all been incorporated in the revised manuscript.

1. Yangi, K. et al. Telemetric intracranial pressure monitoring in patients with hydrocephalus: a systematic literature review. *Front. Pediatr.* 13, 1632216 (2025).
2. Ertl, P., Hermann, E. J., Heissler, H. E. & Krauss, J. K. Telemetric Intracranial Pressure Recording via a Shunt System Integrated Sensor: A Safety and Feasibility Study. *J. Neurol. Surg. Part Cent. Eur. Neurosurg.* 78, 572–575 (2017).

Sara Magnéli, Consultant Neurosurgeon, Uppsala University Hospital, Sweden

Reviewer #2 (Remarks to the Author):

The authors report successful design, fabrication, regulatory approval, and initial placement of pressure monitoring devices into the brains of adult and pediatric patients with hydrocephalus, allowing noninvasive estimation of intracranial pressure and even waveform of the pressure transduction. This is a noteworthy result.

Such a device could significantly improve and make more quantitative the management of possible shunt malfunction and other intracranial pressure-related issues in patients with hydrocephalus. This is not the first device to allow noninvasive measurement of ICP (see, for example, the Radionics Tele-Sensor), but it has the advantage of sensing intraparenchymal pressure, rather than intraluminal shunt pressure within the shunt (which may not adequately reflect true intracranial pressure if there is an obstruction).

The paper gives a strong argument that even "home" measurements of intracranial pressure (ICP) with this device would be potentially useful. However, the illustrations of clinical support for the value of the measurements are somewhat anecdotal. Descriptions are provided of "false alarm" shunt events versus true malfunctions, but even this dichotomy oversimplifies the actual state of shunt management by neurosurgeons. This is probably reasonable in a "proof of concept" presentation, but evaluation of the true value of the measurements to clinicians would depend on a more detailed, presumably Bayesian analysis--potentially beyond the scope of this paper. In the introduction to the main section, there seems to be an implication that this device could resolve issues with diagnosing shunt problems in non-verbal children. I would be very hopeful that this is the case--but evaluation of this question is more nuanced than the clinical histories in the results of the paper.

Author response

We thank the reviewer for their analysis of the potential of the technology. They indicate that the clinical data is "somewhat anecdotal, proof of concept etc". We agree, the main aim of the study was to illustrate the development of the technology and the safety and initial efficacy in a first in human study. Such first in human studies are generally not powered to enable definitive analysis of how the technology may affect patient management. Such a future study would examine the impact of the technology on false alarms, days in hospital, imaging needs etc. We felt it was important to illustrate by examples the type of data that could be obtained. We have not made a change to the manuscript on this point.

More minor comments:

The very low mass of the device is impressive, but the potential risks and problems with the device may depend more on the need for a second penetration of the brain (potential bleeding) and the MRI artifact (loss of diagnostic precision). I found the multiple references to the low mass a bit irrelevant compared to these actual clinical concerns.

Author response

The density of the implant was reported as neurosurgeons would often question us on the potential for migration of the implant through brain tissue and we believed it useful to report the density of the device against other commonly used materials. We accept that the requirement of a second burr hole for sensor placement does introduce new risk and indicated in the revised manuscript "The separate burr hole does present additional risk of bleeding but is part of standard neurosurgical procedures and no complications were noted."

The device appears similar in functionality to the CardioMEMS device (Abbott Laboratories) intended for a different location of pressure measurement (pulmonary vasculature). It appears that this device has a distinct construction, different materials, etc.--yet the claim is made that this device could be used anywhere in the body for pressure measurement (and, potentially, other things.) For a brief announcement of a new technological advance as in this paper, it would be helpful to distinguish from other similar devices available.

Author response

We have added the following text to the revised manuscript;

Clinical interest in pressure sensing has increased over the past decade with the concept of using the pressure readings from the pulmonary artery as an input to titrate drug therapy in heart failure. The Cardiomems device (Abbott Laboratories) might be considered the first generation, the Endotronix device, from Edwards Life Sciences, the second generation and now our fully digital pressure sensor with novel advances in microfabrication to reduce drift and improve fidelity of pressure sensing in general as the next generation.

I am happy to hear of these results and look forward to availability of the device. The

authors should be congratulated for this work!